# Tunable metal-insulator transition, Rashba effect and Weyl Fermions in a relativistic charge-ordered ferroelectric oxide

Jiangang He [1,2], Domenico Di Sante[3], Ronghan Li[4], Xing-Qiu Chen[4], James M. Rondinelli [1] & Cesare Franchini[2]

Controllable metal–insulator transitions (MIT), Rashba–Dresselhaus (RD) spin splitting, and Weyl semimetals are promising schemes for realizing processing devices. Complex oxides are a desirable materials platform for such devices, as they host delicate and tunable charge, spin, orbital, and lattice degrees of freedoms. Here, using first-principles calculations and symmetry analysis, we identify an electric-field tunable MIT, RD effect, and Weyl semimetal in a known, charge-ordered, and polar relativistic oxide $Ag_2BiO_3$ at room temperature. Remarkably, a centrosymmetric $BiO_6$ octahedral-breathing distortion induces a sizable spontaneous ferroelectric polarization through $Bi^{3+}/Bi^{5+}$ charge disproportionation, which stabilizes simultaneously the insulating phase. The continuous attenuation of the $Bi^{3+}/Bi^{5+}$ disproportionation obtained by applying an external electric field reduces the band gap and RD spin splitting and drives the phase transition from a ferroelectric RD insulator to a paraelectric Dirac semimetal, through a topological Weyl semimetal intermediate state. These findings suggest that $Ag_2BiO_3$ is a promising material for spin-orbitonic applications.

[1] Department of Materials Science and Engineering, Northwestern University, Evanston, IL 60208, USA. [2] Faculty of Physics and Center for Computational Materials Science, University of Vienna, Vienna A1080, Austria. [3] Institut für Theoretische Physik und Astrophysik, Universität Würzburg, Am Hubland Campus Süd, Würzburg 97074, Germany. [4] Shenyang National Laboratory for Materials Science, Institute of Metal Research, Chinese Academy of Science, School of Materials Science and Engineering, University of Science and Technology of China, Shenyang 110016 Liaoning, China. Correspondence and requests for materials should be addressed to X.-Q.C. (email: xingqiu.chen@imr.ac.cn) or to J.M.R. (email: jrondinelli@northwestern.edu) or to C.F. (email: cesare.franchini@univie.ac.at)

Cross-controlling of order parameters by external stimuli in a single-phase material is of great interest in both fundamental research and technological applications[1,2]. For instance, metal-to-insulator transitions (MIT) driven by the application of an electric field represent a viable way for designing energy-efficient logic devices[3–5]. Ferroelectric compounds, whose spontaneous electric polarization can be reversed by the application of an external electric field, are among the most promising materials to achieve this type of switch[6], owing to the many different driving forces enabling ferroelectric behavior, including cooperative interactions of lattice distortions, charge, spin, and orbital ordering[7–10]. Among the different types of MIT observed in nature[11] (Mott[12], Peierls[13], Slater[14], Lifshitz[15], Verwey[16], to name a few), the Peierls MIT, associated with a crystal lattice distortion, frequently exhibits a higher transition temperature (e.g., $VO_2$: $T_{MIT} \sim 340$ K;[17] $YNiO_3$: $T_{MIT} \sim 580$ K[18]). Although Peierls-like MITs are ubiquitous in oxides, the breathing mode involving cation-oxygen bond length disproportionation usually does not couple with ferroelectric displacements[17,19–21].

Recent studies have shown that an external electric field could also be employed to switch the Rashba–Dresselhaus (RD) spin splitting in ferroelectric materials[22–24]. The RD spin splitting is a consequence of lifting the spin degeneracy, typically occurring in materials with strong spin–orbit coupling (SOC) lacking inversion symmetry[25–27]. The most interesting outcome of the RD effect is that an electron moving under an electric field (**E**) or a gradient of the crystal potential (**E** = −∇V) behaves as it experiences an effective magnetic field $\mathbf{B}_{eff} \propto \mathbf{E} \times \mathbf{p}/mc^2$, where **p**, $m$, and $c$ are momentum, mass, and speed of light, respectively, that couples to the spin of the electron[28]. Owing to its promising applications in spin-orbitronics devices such as the Datta–Das spin field-effect transistor[29], the RD effect has gained growing research attention[24,28,30–32]. The tunability of the RD splitting by applying an external field is primary possible due to the dependence of $\mathbf{B}_{eff}$ on the crystal potential gradient, which is strongly affected by crystal structure distortions[24,30,32,33].

Additionally, Weyl semimetals are a topological phase whose low energy excitations are the massless chiral fermions[34]. Weyl semimetals are the realization of Weyl fermions in condensed matter systems and exhibit many exotic properties[35]. Since the Weyl semimetal typically only exists in crystals without either time-reversal or inversion symmetry, but not both, it is also possible to activate a Weyl semimetal phase in a topologically trivial material by applying an external field. Although the magnetic field induced Weyl semimetal has been discovered in GdPtBi recently[36], the electric field promoted Weyl semimetal has not been reported yet.

Here, we explore the possibility of simultaneously controlling the MIT, RD spin splitting, and Weyl fermions in a single-phase ferroelectric oxide using an electric field by means of first-principles calculations and symmetry analysis. We demonstrate that this paradigm can be realized in the room temperature phase of the known Peierls-like semiconductor $Ag_2BiO_3$. We identify an atypical polar structural distortion that arises from the octahedral-breathing mode associated with $Bi^{3+}/Bi^{5+}$ charge disproportion. This mechanism enables an unexpected route for tuning the MIT, RD spin splitting, and Weyl semimetallic state simultaneously by applying an external electric field. The modulation of the charge disproportion guides the transition from a polar insulating phase exhibiting RD spin splitting to a nonpolar spin-degenerate Dirac semimetallic state. Remarkably, we find that across the MIT transition there exists an intermediate topological Weyl semimetallic state, manifested by a non-degenerate band crossing around the Fermi level and non-trivial surface states connecting Weyl nodes with opposite chirality.

## Results

**Ground state structural and electronic properties.** At room temperature (and up to at least 380 K) $Ag_2BiO_3$ crystallizes in a polar structure with $Pnn2$ (No. 34) space group[37] (Fig. 1a). At 220 K the polar $Pnn2$ phase is converted to another polar monoclinic structure $Pn$ (No. 7). There is no further observed phase transition down to 2 K[37]. In the $Pnn2$ phase, each octahedra shares an edge (along [100]) and a corner (along [011] and [0$\bar{1}$1] directions) with its adjacent octahedra, resulting in a checkerboard-like distribution of inequivalent $Bi^{3+}/Bi^{5+}$ sites (nominal oxidation state) characterized by different $Bi^{3+}/Bi^{5+}$–O bond lengths of 2.34 and 2.13 Å, respectively[37]. This charge-ordered pattern is responsible for opening a band gap (0.7 eV[38]), which has been observed in other Bi oxides[39,40], and for the onset of ferroelectric behavior, which is observed for the first time. Our density functional theory (DFT) calculations find that the fully relaxed low-temperature $Pn$ phase is nearly degenerate with the $Pnn2$ phase in energy, and correctly reproduce the bond length disproportionation, see Supplementary Tables 1 and 2 for a full structural characterization. We also find a direct band gap of 0.53 eV between the occupied $Bi^{3+}$ and the unoccupied $Bi^{5+}$ 6s orbitals hybridized with O-2p states at the R point **k** = (1/2,1/2,1/2) (Fig. 1c). The slight underestimation of the band gap is due to inaccuracies in the semilocal functional within DFT. The accuracy of DFT in reproducing the crystal structures (lattice constants, volume, and Bi–O bond lengths), energy difference between ferroelectric and paraelectric phases, and band gaps is assessed with various functionals commonly used for ferroelectric oxides[41,42] (Supplementary Table 1).

Since the R point preserves time-reversal symmetry and the whole crystal lacks inversion symmetry, Kramers pairs ($E^{\uparrow}$ (**k**) = $E^{\downarrow}$ (−**k**)) are observed at the conduction band minimum and valence band maximum. Owing to the strong SOC of the Bi cation, both the lowest conduction band and the highest valence band split into two branches, forming inner and outer bands with opposite spin rotation patterns (Fig. 1e, f). The spin splitting ($\Delta E_{RD}$) is larger at the lowest conduction band and exhibits strong anisotropy, manifested by a larger $\Delta E_{RD}$ in the R–T $k_x$ direction (6.4 meV) compared to that in the R–U $k_y$ direction (0.7 meV). Therefore, a combined Rashba and Dresselhaus spin splitting is expected, similar to the case of the formamidinium tin iodide perovskite $FASnI_3$ and consistent with a $C_{2v}$ symmetry, in which both Rashba and Dresselhaus spin splitting are symmetry allowed[33]. The dispersion relation of the resulting electronic states is given, to linear order, by the coupling Hamiltonian $H_{RD} = \alpha_R(k_y\sigma_x - k_x\sigma_y) + \alpha_D(-k_x\sigma_x + k_y\sigma_y)$, with $\alpha_R \sim 0.07$ eV Å and $\alpha_D \sim 0.17$ eV Å the fitted Rashba and Dresselhaus parameters. The predominant contribution of Dresselhaus character, i.e., $\alpha_D > \alpha_R$, is consistent with the calculated spin orientations showing also parallel components to the associated crystal momentum. The ground state band structure of the $Pn$ phase is also insulating and exhibits very similar dispersion (Supplementary Figure 2), consistent with the structural similarity between the $Pnn2$ and $Pn$ phases.

**Symmetry analysis and lattice dynamics.** By means of symmetry analysis based on representation theory[43,44], we find that $Pnn2$ is a subgroup of $Pnna$ (Fig. 1b). Our DFT calculations show that the $Pnna$ phase is only 5.3 meV per atom higher in energy than $Pnn2$, and can be considered as the parent phase of $Pnn2$. The super group of $Pnna$ is $Imma$, which is 1,391 meV per f.u. higher in energy than $Pnn2$. The $Imma$ polymorph is too high in energy to be achieved at elevated temperature. The main structure difference between the $Pnna$ and $Pnn2$ phases is the splitting of the Bi Wyckoff positions in the lower symmetry phase $Pnn2$, which

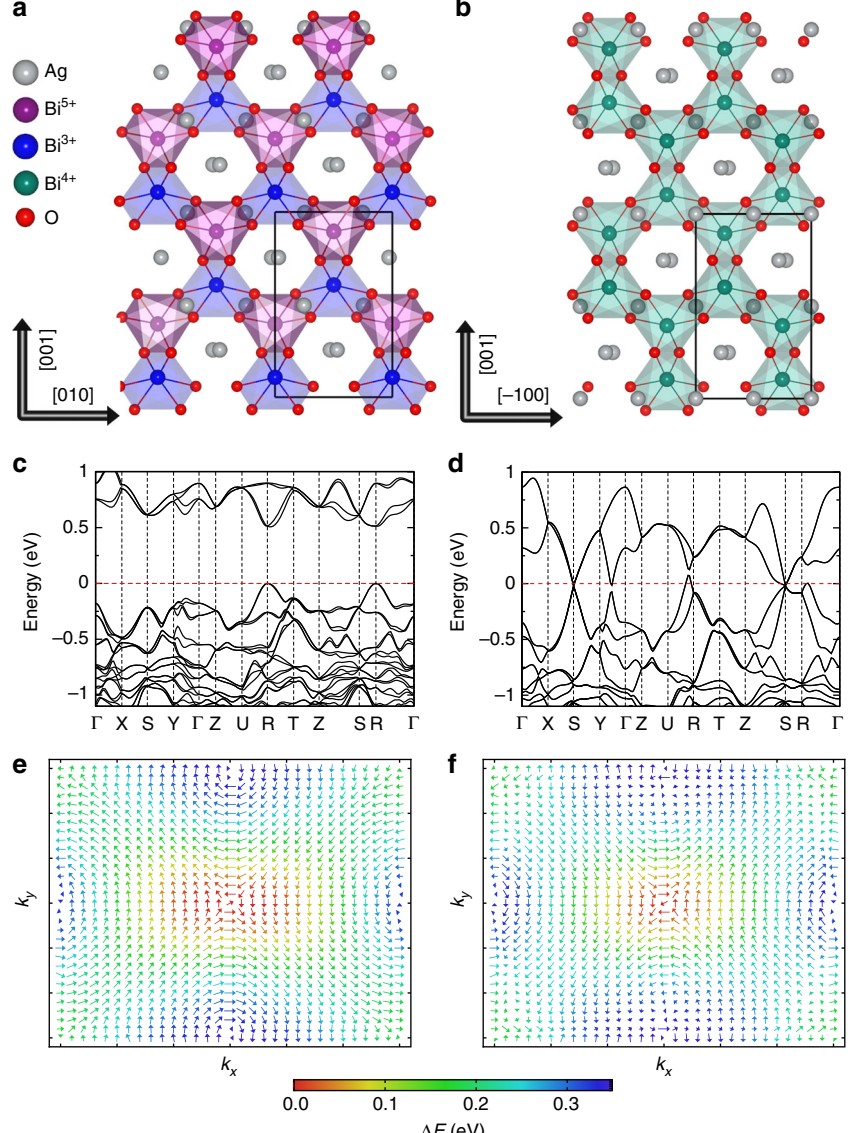

**Fig. 1** Structural and electronic properties of $Ag_2BiO_3$. Crystal structure of **a** the ferroelectric *Pnn*2 and **b** the hypothetical paraelectric *Pnna* phase. Red, gray, green, blue, and purple spheres are $O^{2-}$, $Ag^+$, $Bi^{4+}$, $Bi^{3+}$, and $Bi^{5+}$ ions, respectively. **c** and **d** are the band structures of the *Pnn*2 and *Pnna* phases, respectively. The Fermi level is shifted to 0 eV. High symmetry points in the first Brillouin zone are defined in Supplementary Figure 1. **e** and **f** are the spin textures of the inner and outer branches of conduction bands at the R point in the polar *Pnn*2 phase. The color code indicates the energy level with respect to the bottom of conduction band

permits the $Bi^{3+}/Bi^{5+}$ charge disproportionation (Fig. 1; Supplementary Table 2). From the point view of representation theory, *Pnna* is connected with the room temperature phase *Pnn*2 by a one-dimensional order parameter $\Gamma_{2^-}$ and with the low-temperature phase *Pn* by a combination of the $\Gamma_{2^-}$ and another one-dimensional order parameter $\Gamma_{3^-}$, i.e., $\Gamma_{2^-} \oplus \Gamma_{3^-}$. This assessment is validated by the phonons dispersions of the *Pnna* phase (Fig. 2a), which exhibit two unstable phonon modes at $\Gamma$ with imaginary frequencies $\nu = 392i$ cm$^{-1}$ ($\Gamma_{2^-}$) and $\nu = 30i$ cm$^{-1}$ ($\Gamma_{3^-}$). The condensation of the force constant eigenvector of $\Gamma_{2^-}$ mode directly generates the room temperature phase *Pnn*2, whereas a simultaneous condensation of the $\Gamma_{2^-}$ and $\Gamma_{3^-}$ modes, i.e., $\Gamma_{2^-} \oplus \Gamma_{3^-}$ establishes the *Pn* phase.

The main vibrational characteristic of the $\Gamma_{2^-}$ mode is a Bi–O breathing distortion that causes the $Bi^{3+}/Bi^{5+}$ disproportionation (Fig. 2b, c). Surprisingly, this mode is polar and is the primary order parameter for this paraelectric (*Pnna*) to ferroelectric (*Pnn*2) transition, which produces a spontaneous polarization

($P_z$) of 8.87 μC cm$^{-2}$ (Fig. 3f). Considering that $\Gamma_{2^-}$ is a one-dimensional order parameter and the fact that the calculated polar displacement $Q$ and $\Delta E/\mu$ (where $\Delta E$ and $\mu$ are the energy differences between the paraelectric and ferroelectric states of the material and the dipole moment of ferroelectric phase, respectively) of $Ag_2BiO_3$ are comparable with known ferroelectric materials (Supplementary Figure 3), $Ag_2BiO_3$ should be electric-field tunable with moderate fields.

To the best of our knowledge, this is the first example of a polar octahedra-breathing mode. Typically, the breathing mode in perovskites with the corner-sharing octahedra does not lift inversion symmetry[19]. The reason for such an unusual behavior in $Ag_2BiO_3$ is the coexistence of both corner-sharing and edge-sharing octahedra, which leads to a distorted octahedral framework with inversion centers on occupied Bi cation sites. The bond-disproportionation in combination with the low-crystal symmetry enables an acentric octahedral-breathing mode. As a result, oxygen displacements along the *c* axis in the $BiO_6$

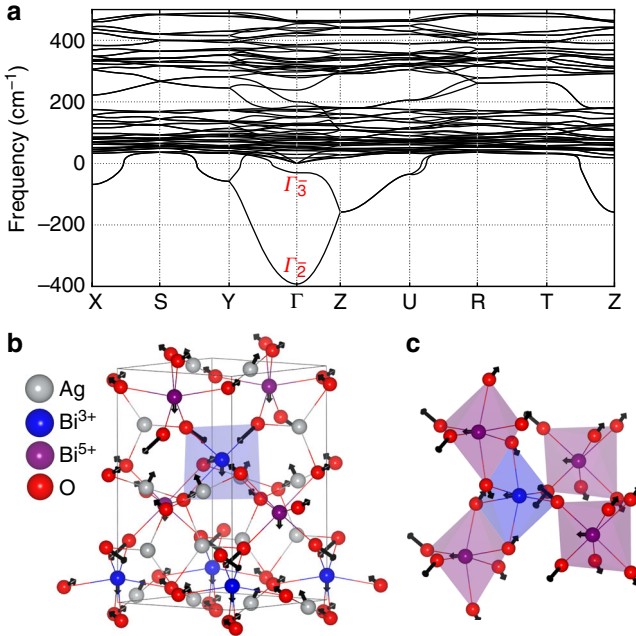

**Fig. 2** Phonon properties of *Pnna*-structured Ag$_2$BiO$_3$. **a** Phonon dispersions of the hypothetical paraelectric phase *Pnna*. Negative frequencies correspond to imaginary modes. **b**, **c** The atomic displacements (force constant eigenvectors, black arrows) of the $\Gamma_{2-}$ mode in real space for BiO$_6$ octahedra and Ag cations, respectively. Note that the amplitude of Ag and Bi cation are enlarged for clarity. The color coding for atoms matches that in Fig. 1

octahedra are not completely compensated, as illustrated in Fig. 2c. Both Ag and Bi atoms have small polar distortions along the *c* axis as well. The polarity of the $\Gamma_{2-}$ mode and its connection with the Bi$^{3+}$/Bi$^{5+}$ disproportionation is the crucial factor enabling the tunability of the electronic structure and spin properties of Ag$_2$BiO$_3$, as elaborated below.

**Tunable MIT with RD and Weyl states**. Unlike the ferroelectric *Pnn*2 phase, the nonpolar *Pnna* phase exhibits a semimetallic band structure without RD spin splitting (Fig. 1d). Nonetheless we find that the *Pnna* phase exhibits a Dirac point at the S point of the Brillouin zone, similar in nature to those defined in double Dirac semimetals[45–47]. This feature is also independent of DFT functional (Supplementary Figure 6). The $\Gamma_{2-}$ mode breaks the mirror symmetry protecting the Dirac point and leads to the opening of a gap, as evidenced in Fig. 4a. This is due to the suppression of the charge disproportionation which produces half-filled Bi$^{4+}$ 6s orbitals crossing the Fermi level (Fig. 1d). This means that the $\Gamma_{2-}$ order parameter simultaneously drives a ferroelectric and a metal-to-insulator transition (Fig. 3). The $\Gamma_{2-}$-driven paraelectric-to-ferroelectric transition is associated with a continuous increase of the spontaneous polarization and of the overall ferroelectric energy gain $\Delta E$ (Fig. 3d, f). The progressive increase of the amplitude of $\Gamma_{2-}$ is coupled with an enhanced charge disproportionation between the two inequivalent Bi sites, measured in terms of Bi–O bond length difference (Fig. 3b), valence Bader charges (Fig. 3c), and by a monotonous increase of the band gap (Fig. 3a). This type of MIT is common to other bismutathes[19,48]. The degree of the ferroelectric distortion influences the RD splitting as well, owing to the coupling between the polar distortion and the potential gradient surrounding the Bi$^{3+}$ and Bi$^{5+}$ cations. This can be quantified by the RD spin splitting coefficient $\alpha_{RD}$ defined as $2\Delta E_{RD}/\Delta k_{RD}$. Figure 3e

illustrates that $\alpha_{RD}$ decreases linearly with the amplitude of $\Gamma_{2-}$ until $Q_{\Gamma_{2-}} \approx 0.25$ Å. For $Q_{\Gamma_{2-}} <0.25$ Å there is a band crossing between the valence and conduction RD bands (Fig. 3g), and then $\alpha_{RD}$ is ill-defined.

Interestingly, our calculations reveal that during the ferroelectric-to-paraelectric transition at $Q_{\Gamma_{2-}} = 0.22$ Å, there exists an intermediate phase that shows the typical hallmark of a Weyl semimetal state, similar to the Weyl semimetal recently found in HgPbO$_3$[49]. Weyl semimetals are a class of quantum materials characterized by nonzero Fermi surface Chern numbers, which manifest as a linear band crossing around the Fermi level with non-degenerate spins in a system with either time-reversal or inversion symmetry broken[50]. In this intermediate phase of Ag$_2$BiO$_3$, the Weyl node is located near the R point with **k** = (0.4975,0.4725,0.4988), about 0.08 eV above the Fermi level (Fig. 4a, b). Symmetry considerations indicate that there are four pairs of Weyl nodes, whose coordinates are given in Supplementary Table 3. The four pairs of Weyl points are protected by a mirror operation, which is preserved through the entire transition from the insulating-ferroelectric phase to the metallic centrosymmetric phase, i.e., always coexisting with the $\Gamma_{2-}$ mode. The Weyl nodes appear only at a specific interval of the $\Gamma_{2-}$ distortions as a result of balancing two competing interactions: (a) going towards the insulating phase the chemical bonding become progressively stronger (bonding/antibonding interactions increase) and destroys the Weyl nodes; (b) on the other side, approaching the metallic phase the broken inversion symmetry gradually fades away, which again results in the a disappearance of the Weyl nodes.

The presence of a Weyl node in the bulk phase is typically associated with topological non-trivial surface states, which connect two Weyl nodes with opposite chirality, and appear as broken Fermi arcs states. To inspect this feature, we have derived the (001) surface electron structure by the Green's function based tight-binding method based on the maximally localized Wannier functions[51–55]. The resulting Fermi arcs presented in Fig. 4b. Moreover, the calculation of the Berry curvature confirms the topological Weyl nature of this intermediate phase (Fig. 4c).

## Discussion

We have established that two pairs of distinct properties, insulator/metal and RD/non-RD, through a Weyl semimetal state, are closely connected with the ferroelectric structure distortion in Ag$_2$BiO$_2$, indicating a potentially tunable MIT and spin splitting by applying an electric field. Since the energy difference between the semimetal paraelectric (*Pnna*) and insulating-ferroelectric (*Pnn*2) phases is only 5.3 meV per atom and the polarization is relatively large (8.87 μC cm$^{-2}$), it is expected that the insulating phase could be suppressed (but not passed) by a relative small electric field (**E**) applied opposite to the direction of the spontaneous polarization (**P**) through the interaction **P** · **E** in the free energy. This is depicted in Fig. 3h based on a Landau–Ginzburg phenomenological model[56]. If the frequency of the electric field pulse is carefully chosen, there should be an oscillation between the *Pnn*2 and *Pnna* phases. By increasing the strength of the electric field, the insulating phase of Ag$_2$BiO$_3$ becomes more conducting, as shown by the density of states provided in Fig. 3a. Since the paraelectric phase is a poor metal, however, the free carriers will tend to screen the external electric field and therefore, the metallic phase is expected to survive only for a short time. This causes the so-called oscillating electroresistance effect (ER), which has potential applications in random access information storage[57]. If the applied pulse electric field is strong enough, however, the Bi$^{3+}$/Bi$^{5+}$ charge-order can be melted and converted

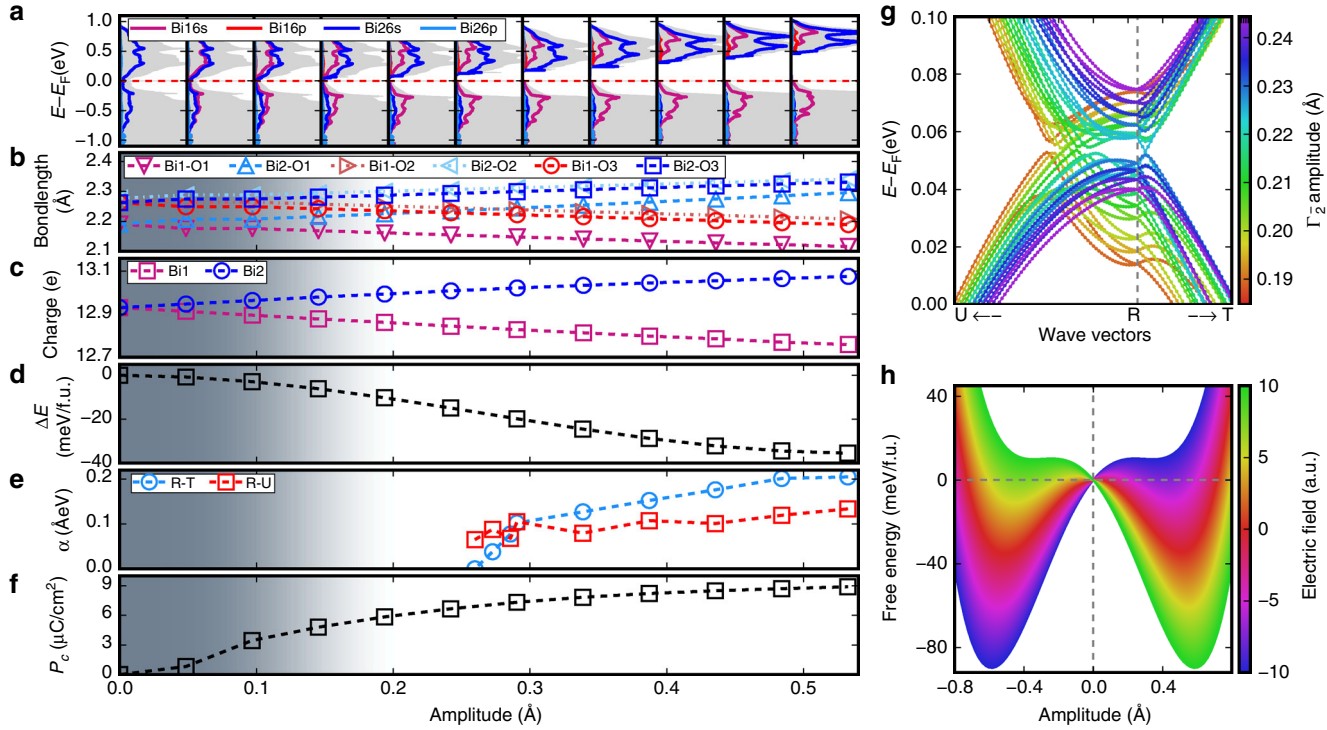

**Fig. 3** The evolution of crystal structure and electronic properties of $Ag_2BiO_3$ as a function of polar mode ($\Gamma_2^-$) amplitude. **a** Projected density of states (the gray shading represents the total density of states), **b** Bi–O bond lengths, **c** Bader charges of Bi atoms (note that since the description of charge states in solid is ambiguous, the charge calculated using Bader's method should not be used for quantitative comparison with the formal charge ($Bi^{3+}$ and $Bi^{5+}$)), **d** energy difference ($\Delta E$) between paraelectric phase (*Pnna*) and ferroelectric phase (*Pnn2*), **e** Rashba–Dresselhaus parameters $\alpha$ along R–U and R–T directions, **f** spontaneous ferroelectric polarization ($P_z$), and **g** band structure along the U–R–T direction. The gray-white gradient background represent the transition from the metallic (gray) and insulating (white) phases. **h** Free energy of ferroelectric $Ag_2BiO_3$ (*Pnn2*) as a function of the external electric field (|**E**|) and polar distortion ($Q_{\Gamma_2^-}$). The free energy density $G$ in the Landau-Ginzburg polynomial expansion with the amplitude of polar distortion $Q_{\Gamma_2^-}$ is: $G = F_0 + \frac{\alpha}{2}Q_{\Gamma_2^-}^2 + \frac{\beta}{4}Q_{\Gamma_2^-}^4 + \frac{\gamma}{6}Q_{\Gamma_2^-}^6 - |\mathbf{E}|Q_{\Gamma_2^-}$, where $F_0$ is the free energy density of the paraelectric phase (*Pnna*). $\alpha$, $\beta$, and $\gamma$ are the coefficients

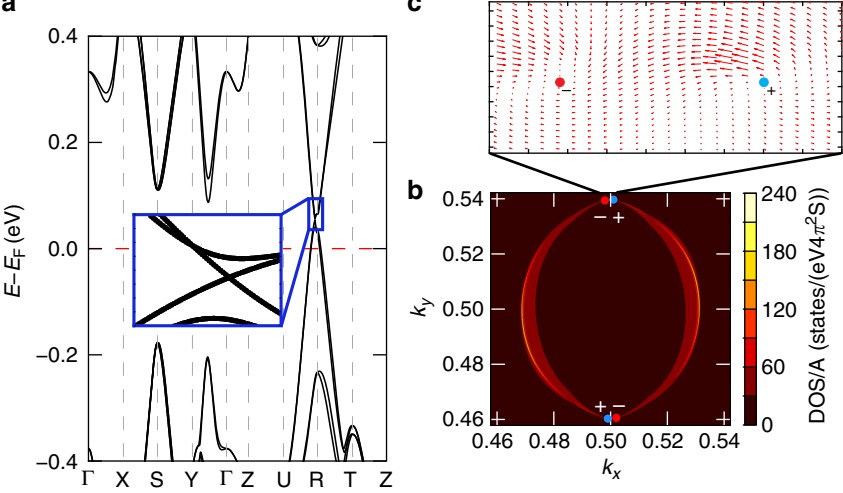

**Fig. 4** Weyl semimetal phase of $Ag_2BiO_3$. **a** Electronic band structure. **b** The corresponding Fermi surface (0.08 eV above the Fermi level) on the (001) surface. The filled circles indicate the projected Weyl points on the surface for two pairs of Weyl points with opposite chirality (+ and −). **c** Two dimensional plot of the Berry curvature showing the opposite chirality of one pair of Weyl nodes. A 3D plot is given in Supplementary Figure 4

into a metallic phase, a phenomena called colossal ER, which has been observed in charge-ordered insulators with strongly competing metallic phases, $Pr_{0.7}Ca_{0.3}MnO_3$, $R_{0.5}Ca_{0.5}MnO_3$ (R = Nd, Gd, and Y)[1,2], and $LuFe_2O_4$[6].

We have demonstrated a controllable metal–insulator transition, Rashba–Dresselhaus spin splitting effect, and Weyl semimetallic state by applying an electric field in the experimentally available charge-ordered oxide $Ag_2BiO_3$ by means of first-principles calculations. The functionalized electric-field tunable MIT could make $Ag_2BiO_3$ a suitable material for lower power memory applications compared to magnetic field controlled MITs. Moreover, the semiconducting nature of $Ag_2BiO_3$

could allow for the injection of a reasonable number of carriers (if the Fermi level is properly tuned) without suppressing the ferroelectric instability, eventually resulting in electrically controllable spin polarized currents as a consequence of the Rashba–Dresselhaus effect[58]. The higher stability of oxides in atmospheric conditions and the higher ferroelectric Curie temperature of $Ag_2BiO_3$ over organic–inorganic metal halide perovskites and GeTe, the prototypical ferroelectric Rashba semiconductors[22,33], make this bismuthate more practical for realistic applications[59]. Finally, the electric-field tunable Weyl semimetallic state proposed here will enable future studies on intertwined ferroelectric and topological phenomena.

## Methods

**Computational details**. All the first-principles calculations were conducted by using the projector-augmented wave pseudopotential method[60,61] as implemented in the Vienna ab intio Simulation Package (VASP)[62,63]. PBEsol exchange-correlation functional[64] and the plane-wave basis set with energy cutoff of 520 eV were used. The Monkhorst-Pack $k$-points grids of $10\times10\times6$, was used to sample the Brillouin zones. All the crystal structures were fully relaxed until the Hellmann–Feynman foreces acting on each atom were less than 0.01 eV Å$^{-1}$. The phonon dispersion was calculated by using finite displacement method as implemented in the Phonopy code[65]. Spin orbital coupling is included in all the electronic structures calculations. The symmetry analysis was conducted using the PSEUDO program provided by the Bilbao crystallographic server[43,66]. The spontaneous ferroelectric polarization was calculated by using Born effective charges ($Z^*$) of the ferroelectric phase ($Pnn2$) and the structure distortion ($u$) of ferroelectric phase with respect to reference paraelectric phase ($Pnna$) as $P_\alpha = \frac{e}{\Omega}\sum_{k,\beta} Z^*_{k,\alpha\beta} u_{k,\beta}$ where, $\Omega$ and $e$ are the volume of unit cell and elementary charge, respectively. The spin texture of the lowest conduction band have been computed by plotting, for each momentum vector on the $k_x$–$k_y$ plane, the expectation values of the Pauli $\sigma$-matrices onto the Kohn–Sham wavefunctions, i.e., the vectorial quantity $S_i(n,k) = \langle\Psi_{n,k}|\sigma_i|\Psi_{n,k}\rangle$, with $i = x,y,z$ and $n$ referring to either the inner or the outer branch of the conduction bands. Represented as a vector, the k-space distribution of the $S_i(n,k)$ resulted in a combined Rashba and Dresselhaus spin-pattern.

The surface state calculations have been performed using a Green's function based tight-binding (TB) approach[51]. The TB model Hamiltonian was constructed by means of maximally localized Wannier functions (MLWFs[52,53]) obtained by the the Wannier90 code[54] and constructed from Bi 5s and O 6p orbitals by employing VASP2WANNIER90[55]. The TB parameters were obtained from the MLWFs overlap matrix. Finally, the berry curvatures based on the first-principles Bloch functions provided by VASP following the recipe described in ref. [67]. The topological charge of each Weyl point (WP) is defined by the integration of the Berry curvature over a closed surface enclosing that WP, and was computed by employing the Wilson-loop method[68].

**Data and code availabty**. All data are available from the corresponding authors upon reasonable request. All codes used in this work are either publicly available or available from the authors upon reasonable request.

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

## Acknowledgments

Work at the University of Vienna was sponsored by the FWF project INDOX (Grant No. I1490-N19). Work at the Shenyang National Laboratory for Materials Science was supported by the National Science Fund for Distinguished Young Scholars (No. 51725103), by the National Natural Science Foundation of China (Grant Nos. 51671193 and 51474202), and by the Science Challenging Project No. TZ2016004. D.D.S. was supported by the German Research Foundation (DFG- SFB 1170) and acknowledges the ERC-StG-336012-Thomale-TOPOLECTRICS. J.M.R. was supported by the Army Research Office (W911NF-15-1-0017). All calculations were performed on the Vienna Scientific Cluster (VSC) and partially at the high-performance computational cluster in the Shenyang National University Science and Technology Park, as well as the National Supercomputing Center in Guangzhou (TH-2 system).

## Author contributions

J.H. and C.F. conceived and coordinated the project. J.M.R. and J.H. performed symmetry analysis of the ferroelectric phase transition. J.H. conducted the DFT calculations. D.D.S. carried out Rashba/Dresselhaus analysis. R.L. and X.-Q.C. analyzed and computed the topological properties (Weyl features) and Berry curvatures. All the authors contributed to discussions and writing of the manuscript.

## Additional information

**Competing interests:** The authors declare no competing financial interests.

