## [Peer Review File · Nature Communications]

Reviewers' comments:

Reviewer #1 (Remarks to the Author):

In this theoretical study, the authors perform density functional theory (DFT) calculations of Ag_2BiO_6 and predict an electric field-induced transition from a polar insulator (Pnn2) to a nonpolar semimetal (Pnna) via an intermediate topological Weyl semimetal state. Starting from its known room temperature polar Pnn2 phase, the authors identify the nonpolar parent group of Ag_2BiO_6 's polar Pnn2 phase – Pnna, which is not yet reported experimentally – and gradually freeze in the polar distortion that connects the two structures. Pnna is predicted, by the DFT calculations, to be a nonpolar spin-degenerate semimetal. The polar nature of Pnn2 results from the charge disproportionation of the $\text{Bi}^{3+}/\text{Bi}^{5+}$ caused by uncompensated octahedral breathing.

By incrementally altering the amplitude of the polar distortion connecting Pnn2 to Pnna, the authors simultaneously tune the charge disproportionation and resulting polarization, the Rashba-Dresselhaus splitting, and the DFT band gap of the material. Interestingly, they find an intermediate Weyl semimetallic phase for a fraction of the full polar distortion. The authors conclude by calculating the Gibbs free energy of the Pnn2 phase as a function of electric field and polar distortion that this could be driven by an external electric field.

The ideas reported are certainly of broad interest: they propose a novel route to the engineering of topological and ferroelectric phases by means of an electric field. However, I am not entirely convinced that this is relevant to experiments, and the manuscript leaves several other questions unaddressed. I detail these questions and concerns below:

1. Is there literature showing hysteresis loops or any switchable polarization for Ag_2BiO_6 ? Similarly, have there been reports of a Pnna metallic phase? While Ag_2BiO_6 is certainly polar, can one label it unambiguously as a ferroelectric? I am concerned that the authors have not definitively established Ag_2BiO_6 as a ferroelectric. If it is ferroelectric, are they certain that Pnna is the most suitable reference phase? Could there be other energetically-competitive nonpolar reference phases (other than Pnna) that are insulating?

From a calculation point of view, since the Pnna phase is metallic, the change in polarization over a structural switching path that includes this phase is ill-defined. Also, Pnna may be predicted to be metallic with DFT-PBE, but we know DFT is not a rigorous theory of the band gap, and PBE in particular tends to underestimate band gaps (see below).

2. Regarding the Pnna phase discussed in the paper, a full investigation of the topological phases in the reported band structures is lacking. In Figure SI (a), the band structure of the Pnna phase is shown. While the intermediate band structure (Fig. 4(a)) report Weyl fermions at the R point, there appears to be several interesting crossings also present in the Pnna phase. For example, the S point appears to host an eightfold degenerate nodal point, which is possibly protected by the crystalline symmetry at the S point (see Phys. Rev. Lett. 116, 186402 (2016)). There are possible crossings between the Y and Gamma point and again between the U and R point in the Pnna phase, which is not clear from the grid sampling used.

3. The authors use the PBEsol exchange-correlation functional for all of their calculations. However, they do not mention the influence of the exchange-correlation functional on the relative energies of the competing structural ground states nor on the $\text{Bi}^{3+}/\text{Bi}^{5+}$ -O bond lengths, which characterize the charge disproportionation and the resulting spontaneous polarization. More importantly, the authors report of band gap of 0.53 eV in the Pnn2 phase and do not address the inadequacies of DFT (and particularly the PBEsol functional) for band gaps. Indeed since the Weyl semimetal phase is induced by closing the gap between the

$\text{Bi}^{3+}/\text{Bi}^{5+}$ states when the charge disproportionation is removed, the Weyl crossing might be robust to calculation method; nevertheless, how sensitive is the electronic structure to functional or the limitations of DFT more broadly?

Aside from these two main points, there are several minor considerations that would clarify and strengthen the work:

- a) Some of the calculation details are missing. The authors should mention which calculations included spin-orbit coupling (SOC); SOC is not mentioned at all in the results nor in the calculation details. Furthermore, the paper should include some details on how the spin textures (Figure 1 (e), (f)) and the chirality of the Weyl points (Table II) were calculated.
- b) The description of the crystal structure differences between the Pn phase and the Pnn2 phase could be elaborated on. In particular, what is the difference in the $\text{Bi}^{3+}/\text{Bi}^{5+}-\text{O}$ bond lengths in these two space groups? The authors report $\text{Bi}^{3+}/\text{Bi}^{5+}-\text{O}$ bond lengths of 2.34 Å and 2.13 Å in the Pnn2 phase, which compare well with experiment; is this comparison made with experiments in the Pnn2 or Pn phase?
- c) An energy difference of 2 meV/f.u. between the Pn and Pnna phase is calculated, which is very small. Could the authors relate the almost-degenerate energies of these two space groups to the particular structural differences in the Pn and Pnna phases, and justify why this energy difference is so small? It would also be interesting to know whether this is dependent on functional used or the final relaxed volume.
- d) It would be useful if the authors commented on what symmetries protect the Weyl crossing at the intermediate value of polar distortion, and how this symmetry change in tuning through the phase transition.
- e) I cannot see how the reference to Table I at the end of the first paragraph in the Results section is relevant to the discussion of the bandgap at the R point.

Reviewer #2 (Remarks to the Author):

The authors in the present manuscript proposed an electric-field switchable metal-insulator transition (MIT) in the ferroelectric oxide Ag_2BiO_3 . Through reducing the $\text{Bi}^{3+}/\text{Bi}^{5+}$ charge disproportionation and ferroelectric polarization, an electric field applied opposite to the direction of spontaneous polarization can drive the phase transition from a ferroelectric Rashba-Dresselhaus (RD) insulator to a paraelectric semimetal. During this process, there appears a Weyl semimetal intermediate state, which hosts multiple Weyl nodes near the R point in BZ.

I think this work is interesting, because it provides the possibility of simultaneously controlling MIT, RD spin splitting and Weyl fermions in ferroelectric oxide by an electric field. However, I have some questions about this work as stated in the following. I think all the questions should be fully addressed before this manuscript can be considered to publish in Nature Communications.

(1) The authors argued that the phase transition from a ferroelectric Rashba-Dresselhaus insulator to a paraelectric semimetal can be obtained by applying an external electric field. They should clearly evaluate the experimental feasibility, e.g., the needed value of electric field. The only comparison of energy difference between these two phases is not enough, considering the possible high transition potential barrier.

(2) I am a bit confused that the states dominating CBM and VBM are s orbitals of Bi, right? Why did the authors construct the TB model Hamiltonian from Bi 6p orbitals by means of Wannier

functions?

(3) In the Weyl semimetal intermediate phase of Ag_2BiO_3 , the authors stated that there are eight pairs of Weyl nodes, and listed their coordinates in Table II of SM. From Table II, however, there are actually four pairs of Weyl nodes, because the coordinates of latter eight Weyl nodes in the table are exactly the same as those of front eight Weyl nodes through translating a reciprocal lattice vector. For example, the position of W8 in the eighth line of Table II is the same as that of W1 in the ninth line. As is known, two Weyl nodes with opposite chirality cannot be in the same position, otherwise they will collapse. So I think there should be four pairs of Weyl nodes in BZ. In addition, I have doubts about the chirality (+ or -) of all eight Weyl nodes shown in the table, based on the symmetry. The authors should double check them. Meanwhile, they should also provide the figure showing the 3D berry curvature near the Weyl node to support the chirality value of it.

(4) There are also some minor errors in the manuscript. For instance, the caption of Figure 1 seems incomplete, and the description of (c, d) is missing. In the last sentence of the first paragraph of Page 4, it should be "Fig. 4(b)", not "Fig. 4(c)".

REPLY TO THE REVIEWER #1

In this theoretical study, the authors perform density functional theory (DFT) calculations of Ag_2BiO_3 and predict an electric field-induced transition from a polar insulator (Pnn2) to a nonpolar semimetal (Pnna) via an intermediate topological Weyl semimetal state. Starting from its known room temperature polar Pnn2 phase, the authors identify the nonpolar parent group of Ag_2BiO_3 's polar Pnn2 phase Pnna, which is not yet reported experimentally and gradually freeze in the polar distortion that connects the two structures. Pnna is predicted, by the DFT calculations, to be a nonpolar spin-degenerate semimetal. The polar nature of Pnn2 results from the charge disproportionation of the $\text{Bi}^{3+}/\text{Bi}^{5+}$ caused by uncompensated octahedral breathing.

By incrementally altering the amplitude of the polar distortion connecting Pnn2 to Pnna, the authors simultaneously tune the charge disproportionation and resulting polarization, the Rashba-Dresselhaus splitting, and the DFT band gap of the material. Interestingly, they find an intermediate Weyl semimetallic phase for a fraction of the full polar distortion. The authors conclude by calculating the Gibbs free energy of the Pnn2 phase as a function of electric field and polar distortion that this could be driven by an external electric field.

The ideas reported are certainly of broad interest: they propose a novel route to the engineering of topological and ferroelectric phases by means of an electric field. However, I am not entirely convinced that this is relevant to experiments, and the manuscript leaves several other questions unaddressed. I detail these questions and concerns below:

- We thank the referee for finding our work of broad interest.

1. Is there literature showing hysteresis loops or any switchable polarization for Ag_2BiO_3 ? Similarly, have there been reports of a Pnna metallic phase?

While Ag_2BiO_3 is certainly polar, can one label it unambiguously as a ferroelectric?

I am concerned that the authors have not definitively established Ag_2BiO_3 as a ferroelectric.

If it is ferroelectric, are they certain that Pnna is the most suitable reference phase? Could there be other energetically-competitive nonpolar reference phases (other than Pnna) that are insulating?

From a calculation point of view, since the Pnna phase is metallic, the change in polarization over a structural switching path that includes this phase is ill-defined.

Also, Pnna may be predicted to be metallic with DFT-PBE, but we know DFT is not a rigorous theory of the band gap, and PBE in particular tends to underestimate band gaps (see below).

We address the reviewer's questions one-by-one in the order raised below:

- As far as we know, there are no literature reports on electrical poling of this material, which usually demands high quality and large bulk crystals. We hope our research will encourage experimentalists to both synthesize such samples and perform ferroelectric polarization-field measurements.

- Theoretically there is no way to unambiguously prove the material is ferroelectric, because the ferroic assignment of a spontaneous and reversible polarization can only be assessed via experimentation; however, from a computational perspective we can assess quantities for this material and compare values for Ag_2BiO_3 to known ferroelectrics. For example, two important quantities are $\Delta E/\mu$ (where ΔE and μ are the energy differences between the paraelectric and ferroelectric states of the material and the dipole moment of the ferroelectric phase, respectively) and the polar distortion amplitude (Q). The latter of these quantities was formulated by Abrahams et. al. (Phys. Rev. 172, 551) and is used to predict successfully ferroelectric to paraelectric phase transition temperatures. The calculated $\Delta E/\mu$ and Q for well-known ferroelectric compounds and Ag_2BiO_3 are now reported in Fig. S5 of the Supplementary Information. The values clearly indicate that Ag_2BiO_3 should be switchable based on these criteria. We have added the corresponding discussion in the revised version.

- The $Pnna$ phase was firstly reported by Deiele and Jansen in 1999 (see J. Solid State Chem. 147, 117). Owing to the inconsistencies between the Bi^{4+} chemical assignment and the semiconducting behavior, they re-investigated Ag_2BiO_3 with new x-ray diffraction, neutron powder diffraction, and nonlinear optical measurements at different temperatures. Their new finding shows that the space group for Ag_2BiO_3 between 373K and 298K is $Pnn2$ (see Solid State Science 8, 267). This finding indicates that the structural difference between $Pnn2$ and $Pnna$ is very small (change in a glide operation, X-ray diffraction could not distinguish it very well) and $Pnna$ is likely to be the closest paraelectric phase. As stated in our paper, we also found the centrosymmetric $Pnna$ as a supergroup of $Pnn2$ according to the International Tables of Crystallography and is also supported by our phonon calculations. We have performed a careful supergroup search based on group theory analysis and considered other possible supergroups. Our group theory analysis and phonon calculation confirm that $Pnna$ is the best, lowest energy structure of the paraelectric phase.

- We did not intend to indicate that the switching procedure would occur in the "poor" metallic $Pnna$ phase. We proposed to use electric field pulses applied in a direction opposing the polarization in the ferroelectric phase to reduce the magnitude of the polarization. As the material approaches the $Pnna$ phase, the external electric field will be better screened and it is likely to go back to the $Pnn2$ ground state of the same parity. If the frequency of the electric field pulse is carefully chosen, there should be an oscillation between the $Pnn2$ and $Pnna$ phases. During this dynamic process, the MIT and topological phase transition will occur. We have revised our paper accordingly to make this point clearer. We also change the "Switchable" in the title of our manuscript to "Tunable" to better represent our results.

- If the oxidization state of Bi is +4, the compound is definitely a metal or semimetal if there is no magnetic moment because the $6s^1$ band is half filled and no gap can appear without a lattice distortion. To further confirm our result, we calculated the band structure and DOS of Ag_2BiO_3 using the screened hybrid exchange-correlation functional HSE06 as

well as meta-GGA functional SCAN (Phys. Rev. Lett. 115, 036402), including spin-orbital coupling (SOC). Similar to PBEsol, LDA, PBE, SCAN, and HSE06 indicate that the $Pnna$ phase of Ag_2BiO_3 is a semimetal with a symmetry predicted band crossing at R point (see Supplementary Information, Figs. S2). Since the point on the Fermi surface is protected it cannot be lifted without a lattice distortion, which would then change the nominal oxidation state of 4+ for the two bismuth sites, i.e., 3+ and 5+, turning the systems into an insulating regime. Also, it is worth noting that HSE06 tends to overestimate the band gap of small band gap compounds (for example, see J. Phys. Condens. Matter, 2017 and J. Phys. Condens. Matter, 2008). We also added discussion on this point in the revised manuscript.

2. Regarding the $Pnna$ phase discussed in the paper, a full investigation of the topological phases in the reported band structures is lacking. In Figure SI (a), the band structure of the $Pnna$ phase is shown. While the intermediate band structure (Fig. 4(a)) report Weyl fermions at the R point, there appears to be several interesting crossings also present in the $Pnna$ phase. For example, the S point appears to host an eightfold degenerate nodal point, which is possibly protected by the crystalline symmetry at the S point (see Phys. Rev. Lett. 116, 186402 (2016)). There are possible crossings between the Y and Γ point and again between the U and R point in the $Pnna$ phase, which is not clear from the grid sampling used.

We thank the referee for raising this question. We have double-checked our analysis of possible crossing points in the $Pnna$ phase, i.e., the S point as well as along the U-R and Y- Γ directions, by increasing the number of K-point along these high symmetry points. We found there are sizable (nonzero) band gaps along the U-R and Y- Γ directions and only S is a symmetry protected Dirac point, similar to the class of Dirac points recently defined in Phys. Rev. Lett. 116, 186402. We have revised our manuscript accordingly.

3. The authors use the PBEsol exchange-correlation functional for all of their calculations. However, they do not mention the influence of the exchange-correlation functional on the relative energies of the competing structural ground states nor on the $\text{Bi}^{3+}/\text{Bi}^{5+}$ -O bond lengths, which characterize the charge disproportionation and the resulting spontaneous polarization. More importantly, the authors report of band gap of 0.53 eV in the $Pnn2$ phase and do not address the inadequacies of DFT (and particularly the PBEsol functional) for band gaps. Indeed since the Weyl semimetal phase is induced by closing the gap between the $\text{Bi}^{3+}/\text{Bi}^{5+}$ states when the charge disproportionation is removed, the Weyl crossing might be robust to calculation method; nevertheless, how sensitive is the electronic structure to functional or the limitations of DFT more broadly?

- The referee is certainly right, in the sense that it is well known that standard DFT tends to underestimate band gaps, in particular for wide band gap insulators. However, for the $Pnn2$ phase PBEsol delivers a band gap of 0.53 eV, which is only slightly smaller than the experimental value, 0.7 eV (J. Solid State Chem. 147, 117 (1999)), which justify the adoption of this functional. Conversely, HSE06 overestimates the band gap significantly, 1.63 eV. Moreover, also our relaxed structures using PBEsol agree very well with the experiment

data, as reported in the Table I of the Supplementary Information. This is not surprising considering that PBEsol is based on the widely used PBE functional and is optimized specifically for periodic system, including bulk and surface (Phys. Rev. Lett. 100, 136406). It performs well for $\text{Bi}^{3+}/\text{Bi}^{5+}$ charge disproportionated systems (for example, Chem. Mater. 29, 2445) and has been widely used for ferroelectric studies (for examples, Adv. Funct. Mater. 23, 4810; Phys. Rev. B 85, 054417; Sci. Rep. 7, 43482; Phys. Rev. B 83, 094105).

In the revised manuscript, we provide structural data, energetics and electronic properties using other functionals: LDA, PBE, SCAN and (to a lesser extent) HSE06. The results are collected in Table I and Figs. S2-S5 of Supplementary Information. We have also revised our manuscript and added the discussion about the influence of exchange-correlation functional.

Aside from these two main points, there are several minor considerations that would clarify and strengthen the work:

a) Some of the calculation details are missing. The authors should mention which calculations included spin-orbit coupling (SOC); SOC is not mentioned at all in the results nor in the calculation details. Furthermore, the paper should include some details on how the spin textures (Figure 1 (e), (f)) and the chirality of the Weyl points (Table II) were calculated.

● We thank the review for pointing this out. We have revised our paper accordingly by adding the required details.

b) The description of the crystal structure differences between the Pn phase and the $Pnn2$ phase could be elaborated on. In particular, what is the difference in the $\text{Bi}^{3+}/\text{Bi}^{5+}\text{-O}$ bond lengths in these two space groups? The authors report $\text{Bi}^{3+}/\text{Bi}^{5+}\text{-O}$ bond lengths of 2.34 Å and 2.13 Å in the $Pnn2$ phase, which compare well with experiment; is this comparison made with experiments in the $Pnn2$ or Pn phase?

● We have added a picture on the structure distortions connecting the $Pnn2$ to Pn polymorphs in the Fig. S1 of the Supplementary Information of the revised version. The symmetry broken (Wyckoff sites splitting) from $Pnn2$ to Pn mainly happens on the Ag and Oxygen sites. The number of unique Bi-O bond lengths within the BiO_6 octahedra in the Pn phase increases to 6 from 3 in the $Pnn2$ phase. We have added Bi-O bond lengths in the Table 1 of the Supplementary Information for these two phases as well. It is worth noting that the computation we made in this work is based on the $Pnn2$ phase and the low-temperature monoclinic Pn structure is not related.

c) An energy difference of 2 meV/f.u. between the Pn and $Pnna$ phase is calculated, which is very small. Could the authors relate the almost-degenerate energies of these two space groups to the particular structural differences in the Pn and $Pnna$ phases, and justify why this energy difference is so small? It would also be interesting to know whether this is dependent on functional used or the final relaxed volume.

● We have carefully checked the energy difference between the $Pnn2$ and Pn phases using different functionals and found they are indeed energetically degenerate for all the

functionals. The main reason of small energy difference is that the structure change from $Pnn2$ to Pn is very small: the monoclinic angle is just slightly away from 90° , i.e., the atom position and cell volumes of the two structures are effectively the same. The crystal class change arises from a spontaneous strain. Similar results have been observed in BiFeO_3 thin films, i.e., the reported Cc phase (Science 326, 977) is basically the rhombohedral $R3c$ phase except that the Cc structure allows for a monoclinic distortion (Phys. Rev. B 83, 094105). Again, the low temperature phase Pn is not related to any of the main results described in the manuscript which focuses rather on the room temperature ferroelectric ($Pnn2$) and high temperature paraelectric ($Pnna$) phases.

d) It would be useful if the authors commented on what symmetries protect the Weyl crossing at the intermediate value of polar distortion, and how this symmetry change in tuning through the phase transition.

We thank the referee for pointing out this interesting issue. The four pairs of Weyl points are protected by a mirror operation. This mirror operation is preserved through the entire transition from the insulating-ferroelectric phase to the metallic centrosymmetric phase, i.e., always coexisting with the Γ_2^- mode. The Weyl nodes appears only at a specific interval of the Γ_2^- -distortions as a result of balancing two competing interactions: (a) going towards the insulating phase the chemical bonding become progressively stronger (bonding/antibonding interactions increase) and destroys the Weyl nodes; (b) on the other side, approaching the metallic phase the broken inversion symmetry gradually fades away, which again results into a disappearance of the Weyl nodes. We have added a comment on this issue in the main text.

e) I cannot see how the reference to Table I at the end of the first paragraph in the Results section is relevant to the discussion of the bandgap at the R point.

- We amended this issue.

REPLY TO THE REVIEWER #2

The authors in the present manuscript proposed an electric-field switchable metal-insulator transition (MIT) in the ferroelectric oxide Ag_2BiO_3 . Through reducing the $\text{Bi}^{3+}/\text{Bi}^{5+}$ charge disproportionation and ferroelectric polarization, an electric field applied opposite to the direction of spontaneous polarization can drive the phase transition from a ferroelectric Rashba-Dresselhaus (RD) insulator to a paraelectric semimetal. During this process, there appears a Weyl semimetal intermediate state, which hosts multiple Weyl nodes near the R point in BZ.

I think this work is interesting, because it provides the possibility of simultaneously controlling MIT, RD spin splitting and Weyl fermions in ferroelectric oxide by an electric field. However, I have some questions about this work as stated in the following. I think all the questions should be fully addressed before this manuscript can be considered to publish in Nature Communications.

- We appreciate the review for his/her nice comments and the recommendation.

(1) The authors argued that the phase transition from a ferroelectric Rashba-Dresselhaus insulator to a paraelectric semimetal can be obtained by applying an external electric field. They should clearly evaluate the experimental feasibility, e.g., the needed value of electric field. The only comparison of energy difference between these two phases is not enough, considering the possible high transition potential barrier.

- As far as we are aware, it is not practical to calculate the coercive switching field by DFT because switching relies on domain wall motion, dielectric screening properties, and temperature. An alternative to performing molecular dynamics or Monte Carlo simulations relies on the widely used approximation for estimating the energy barrier with respect to polarization (or dipole moment). This estimate is based on the fact that a larger energy stabilization can be gained for an electric field coupled with a larger dipole moment. Therefore, the compound with a large polarization (dipole moment) can overcome a higher energy barrier. In reality, however, the poling electric field can be much smaller than this estimation because of the domain structure and the finite temperature effect.

We estimated the electric field (E) required to overcome the energy barrier between ferroelectric and paraelectric phases using the following formula $E \approx \frac{\Delta E}{\mu}$, where ΔE and μ are energy difference between ferroelectric and paraelectric phases and dipole moment, respectively. The results are collected in Figure S5 of the Supplementary Information, where we also provide a comparison with other well-known ferroelectric materials. The estimated electric field required to switch Ag_2BiO_3 is smaller than $\text{RbBiNb}_2\text{O}_7$, comparable with BiFeO_3 and LiNbO_3 , and two times larger than PbTiO_3 with different functionals. This indicates Ag_2BiO_3 is likely switchable with an electric field.

(2) I am a bit confused that the states dominating CBM and VBM are s orbitals of Bi, right? Why did the authors construct the TB model Hamiltonian from Bi 6p orbitals by means of Wannier functions?

We thank the referee for pointing out this discrepancy; the referee is obviously right. We made a typo in reporting the orbital basis used for the wannier projection. For the construction of the Wannier function we have included the Bi-s and the O-p orbitals. We have corrected this mistake in the methods section.

(3) In the Weyl semimetal intermediate phase of Ag_2BiO_3 , the authors stated that there are eight pairs of Weyl nodes, and listed their coordinates in Table II of SM. From Table II, however, there are actually four pairs of Weyl nodes, because the coordinates of latter eight Weyl nodes in the table are exactly the same as those of front eight Weyl nodes

through translating a reciprocal lattice vector. For example, the position of W8 in the eighth line of Table II is the same as that of W1 in the ninth line. As is known, two Weyl nodes with opposite chirality cannot be in the same position, otherwise they will collapse. So I think there should be four pairs of Weyl nodes in BZ. In addition, I have doubts about the chirality (+ or -) of all eight Weyl nodes shown in the table, based on the symmetry. The authors should double check them. Meanwhile, they should also provide the figure showing the 3D berry curvature near the Weyl node to support the chirality value of it.

● We thank the referee for this very important criticism. We admit that we have not correctly listed the Weyl points in Table II of the previous Supplementary Information. As the referee correctly points out, the last eight points in the table are exactly identical with the first eight ones. We have amended this error in the revised version of the manuscript. Also, as recommended by the Referee we have re-analyzed the chirality of all Weyl nodes by computing the Berry curvature. The results obtained by the Berry curvature calculations confirm that the four pairs of crossing points are indeed Weyl points with opposite chirality. In the revised manuscript we have added a 2D (Fig. 4c of the main text) and 3D (Fig. S9 of the Supplementary Information) plot of the Berry curvature around the R point together with a refined picture of the surface Fermi arcs obtained with a much denser mesh.

(4) There are also some minor errors in the manuscript. For instance, the caption of Figure 1 seems incomplete, and the description of (c, d) is missing. In the last sentence of the first paragraph of Page 4, it should be Fig. 4(b), not Fig. 4(c).

● Thanks! We have carefully check our manuscript and corrected all the typos.

REVIEWERS' COMMENTS:

Reviewer #1 (Remarks to the Author):

The authors have responded thoroughly to my comments, and I think the manuscript is much improved. In particular, the comparison of the different functionals GGA, HSE, SCAN, etc. is a useful addition (with the caveat that none of these functionals is a panacea or rigorous for the band gap). The authors now fully characterise the Dirac points in terms of symmetry protection, and discuss the effect of an applied electric field on the metallic phase. All of this strengthens the manuscript considerably.

I recommend publication, although prior to publication, the authors should proof read the manuscript another time to correct typos, and revise the spin texture plots in the SI, which are presented such that the arrows are too small to see their spin directions.

Reviewer #2 (Remarks to the Author):

All the responses to my previous comments are satisfactory. I thus recommend the publication of the paper in Nature Communications.